# Governance and Power Dynamics in a Small-Scale Hilsa Shad (*Tenualosa ilisha*) Fishery: A Case Study from Bangladesh

**Mohammad Mojibul Hoque Mozumder** [1,*] **, Aili Pyhälä** [2] **, Md. Abdul Wahab** [3] **, Simo Sarkki** [4] **, Petra Schneider** [5] **and Mohammad Mahmudul Islam** [6,7]

1    Helsinki Institute of Sustainability Science (HELSUS), Doctoral Programme in Interdisciplinary Environmental Science (DENVI), Faculty of Biological and Environmental Sciences, University of Helsinki, 00014 Helsinki, Finland
2    Department of Geosciences and Geography, Faculty of Social Sciences, Helsinki Institute of Sustainability Science (HELSUS), University of Helsinki, 00014 Helsinki, Finland; aili.pyhala@helsinki.fi
3    WorldFish, Bangladesh, and South Asia Office, Dhaka 1213, Bangladesh; A.Wahab@cgiar.org
4    Cultural Anthropology, Faculty of Humanities, University of Oulu, 90014 Oulu, Finland; simo.sarkki@oulu.fi
5    Department for Water, Environment, Civil Engineering and Safety, University of Applied Sciences Magdeburg-Stendal, D-39114 Magdeburg, Germany; petra.schneider@h2.de
6    Department of Coastal and Marine Fisheries, Sylhet Agricultural University, Sylhet 3100, Bangladesh; mahmud.cmf@sau.ac.bd
7    Department of Geography, Memorial University of Newfoundland, St.John's, NL A1B 3X9, Canada
*    Correspondence: mohammad.mozumder@helsinki.fi; Tel.: +358-400-491-395

**Abstract:** This paper considers the hilsa shad (*Tenualosa ilisha*) fishery of southern Bangladesh as a case study regarding governance and power dynamics at play in a small-scale fishery, and the relevance of these for the sustainable management of coastal fisheries. Qualitative methods, involving in-depth individual interviews (n = 128) and focus group discussions (n = 8) with key stakeholders in the hilsa fishery, were used to capture multiple perspectives on governance from those in different positions in the relative power structures studied, while facilitating insightful discussions and reflections. The analysis here is based on a power cube framework along three power dimensions (levels, spaces, and forms) in Bangladesh's hilsa fishery. The study displays an imbalance in the present hilsa governance structure, with some stakeholders exercising more power than others, sidelining small-scale fishers, and encouraging increasing illegal fishing levels that ultimately harm both the fisheries and those dependent on them. To overcome this, we propose a co-management system that can play a vital role in equalizing power asymmetry among hilsa fishery stakeholders and ensure effective hilsa fishery governance. Our results suggest that recognizing analyzed power dynamics has substantial implications for the planning and implementation of such co-management and the long-term sustainability of the hilsa fishery.

**Keywords:** small-scale fishery; hilsa fishery; governance; power cube; power asymmetry; co-management

## 1. Introduction

Small-scale fisheries (SSFs) are defined in multiple ways. The interchangeable terms generally associated with SSF, such as "artisanal", "local", "coastal", "traditional", "small", "subsistence", "nonindustrial", "low-tech", and "poor", are indicative of the many values and characteristics underpinning the definitions [1]. For some scholars, the term "small-scale fishery" (SSF) evokes a

mental image of small, traditional fishing crafts equipped with low-tech gear requiring labor-intensive fishing methods [2].

SSFs in developing countries are a significant and valuable component of global fisheries [3]. According to recent estimates, 97% (~36 million) of the world's fishers are in developing countries, and 88% (~107 million) of the world's fishery and fish trade workers are employed in the small-scale sector in developing countries [4]. Apart from contributing to global markets, SSFs are also essential for food security in developing countries. However, SSF faces several threats, including overfishing in degraded coastal and riparian ecosystems [5–7].

Several other threats to SSF include competition with industrial fleets, water pollution, destruction of fish habitats, and an increasing human population and demand for land and resources in coastal areas [8]. With this growing demand and pressures such as climate change, lack of financial sustainability, inadequate equipment and infrastructure, and limited access to markets, it becomes ever more critical that SSFs are managed sustainably [9]. In addressing these challenges, there is a lack of adequate fisheries management mechanisms in place [10]. Under these circumstances, the sustainability and economic viability of SSFs are seriously threatened, warranting improved governance [11].

Good governance is a fundamental tenet for sustainable fisheries management [12]. Fisheries governance is the sum of the legal, social, economic, and political arrangements used to manage fisheries [13]. It comprises international, national, and local dimensions, and can include legally binding rules as well as customary social arrangements [14]. The literature has broadly established that the conventional approach (via stock status and top-down driven regulatory measures) alone cannot address the complex socio-economic characteristics, multiple livelihood needs, and the multi-species nature of many SSFs [15]. Several scholars agree that establishing appropriate means of governance is paramount for SSFs to thrive [16–18].

Fisheries management problems are related to power dynamics among stakeholders [19]. Power in capture fisheries is considered an opportunity to participate in, and influence decision making in the management of fisheries resources [20]. Understanding power and power relations are central to discussions of participation and empowerment, with the need to investigate how power is manifested and by whom [21]. Power can be a constructive force, but it can also be disruptive and corruptive and may serve special interests [22]. Thus, power is a relative aspect; it characterizes relationships between individuals or groups. Debates over power relations' impact on fisheries governance underscore issues concerning the quality of life and livelihood opportunities for marginalized riparian communities [23]. To avoid community authority, power relationships should be understood and examined as embedded within a social-ecological system (SES), such as SSF [24]. There is little consensus in the literature about how power dynamics affect linkages between desired outcomes and different forms of governance in SSFs [25,26].

In this study, building on existing research [19,27,28], the power cube framework [29] has been employed for the study of governance and power dynamics in one particular small-scale hilsa shad fishery in Bangladesh (see Section 2 for more details). The research questions of the present study are as follows:

i.　How did the existing systems of governance in the hilsa fishery develop and begin to operate?

ii.　How is power exercised/distributed, and how does it discriminate among the stakeholders in hilsa value chains?

iii.　What spaces (potential arenas for participation and action) and forms of power exist in the fishery? How do they play out among different groups of stakeholders in each of these spaces?

This article concludes by presenting conclusions and recommendations both for policy support and further research.

## 2. Hilsa Fishery in Bangladesh

Bangladesh is one of the world's leading fish-producing countries with a total production of 4.134 million metric tons in 2016–2017, of which the hilsa (*Tenualosa ilisha*) catch makes up approximately 12% [30]. In Bangladesh, fish—including hilsa—come mainly from two sources: inland and marine. Hilsa is an anadromous fish that spends part of its life in the marine ecosystem and part of its life in freshwater rivers. It shows a distinct migration pattern from the Bay of Bengal to the rivers, the Padma, Meghna, and its tributaries for breeding and nursing purposes [31]. Hilsa is also found in the Indian Ocean and the Arabian Sea. The hilsa fishery in Bangladesh has a total annual value of US $1.3 billion, accounting for more than 4.3% of the nation's total GDP and employing approximately 2.5 million people directly and indirectly in the process [32]. Thus, hilsa has become the most valuable single-species fishery of Bangladesh. Hilsa is also crucial to the Bangladeshi diet for its nutritional value, as it is rich in micronutrients and omega-3 fatty acids [33]. The social and cultural significance of hilsa is also immense: hilsa is honored as the national fish of Bangladesh and considered essential in many religious, social, and festive events [34].

Since the 1950s, the Government of Bangladesh (GoB) has passed several acts, ordinances, and rules to provide a framework for exploiting, developing, managing, and conserving its fisheries sector and aquatic resources (Table 1). Also, GoB introduced several projects with different organizations as a partnership for the sustainability of the hilsa fishery (Table 1).

**Table 1.** Evolution of hilsa governance, revised from [35].

| Policy, Plans, Compliances, and Projects | Issues Related to Hilsa Fishery Governance | Year |
|---|---|---|
| The Protection and Conservation of Fish Act (PCFA) | • Fishing nets with a mesh size of less than 4.5 cm are prohibited.<br>• The manufacturing, import, marketing, storing, transportation, and owning and use of monofilament gill nets (Current Jal) are prohibited. | 1950 |
| The Marine Fisheries Ordinance and rules | • Two fishing zones for artisanal and industrial fishing, within and beyond a 40 m depth, are stipulated<br>• Fishing with gear that does not meet specified mesh size, and with any kind of explosives, poisons, or other harmful substances, is prohibited | 1983 |
| The New Fisheries Management Policy | • Addresses the over-exploitation of fishery resources and inequality of fishing rights<br>• Sets objectives for bringing the most significant benefits of all national fisheries to fishers instead of non-fisher elites<br>• Adopts conservation measures to ensure that resources are sustained | 1986 |
| The National Fisheries Policy | • Goal of enhancing fisheries' resources and production<br>• Combating malnutrition by meeting the need for animal protein with fish<br>• Goal of alleviating poverty through creating self-employment and enhancing the socio-economic conditions of fishers<br>• Goal of achieving economic growth and earning foreign currency by exporting fish and fisheries products | 1998 |
| The Hilsa Fisheries Management Action Plan (HFMAP) | • Enforcing compliance with conservation rules and regulations, and strategies related to the hilsa fishery<br>• Supports a sustainable hilsa fishery, protecting critical habitats, and building the capacity of fisheries' actors<br>• Offers alternative livelihoods for jatka fishers based on a compensation scheme<br>• Raises mass awareness of the need for jatka and hilsa conservation | 2003 |

**Table 1.** *Cont.*

| Policy, Plans, Compliances, and Projects | Issues Related to Hilsa Fishery Governance | Year |
|---|---|---|
| Formation of hilsa sanctuaries | • The government declares four areas in the Meghna, Tetulia, and Andharmanik Rivers and some estuarine waters as hilsa sanctuaries. <br>• Altogether six hilsa sanctuaries are established by the government, following the HFMAP. | 2005 |
| The National Fisheries Strategy | • Promotes and supports collaboration, linkages, and partnerships for the benefit of marine fisheries <br>• Promotes the participation of fishers, and other stakeholders in the fisheries' value chain, local communities, the private sector, and NGOs (non-governmental organizations) in government programs through the DoF. | 2006 |
| Jatka conservation | • Provides food compensation to hilsa fisher households <br>• Builds awareness of conservation, supporting alternative income-generating activities (AIGAs) <br>• Imposing regulations to prevent *jatka* and brood hilsa fishing during the ban periods | 2008 |
| Formation of the 5th hilsa sanctuary | • A 20 km stretch of the Padma River's lower basin from Narhira to Bhedarganj, Shariatpur District | 2011 |
| Formation of the 6th hilsa sanctuary | • At the confluence of the Meghna, Arial Kha, Kala Bador, and Kirton Khola Rivers | 2018 |
| International Institute for environment and development (IIED) initiatives | • Innovative ways to tackle overfishing problems and allow threatened hilsa fish stocks in Bangladesh to recover were devised by IIED and partners (Bangladesh Centre for Advanced Studies and Bangladesh Agricultural University) and collaborated with the Department of Fisheries of the government of Bangladesh). <br>• To work with affected communities and ecosystems to learn what is working and what is not and found ways to improve it. | 2013–2016 |
| Enhanced Coastal Fisheries in Bangladesh (ECOFISH-BD) project | • Fortifies science-based decision-making in the hilsa fishery and its aquatic ecosystem <br>• Steers adaptive co-management in the sanctuaries <br>• Enhances the socio-ecological and economic resilience of fishing households and communities through improving policy, power, and incentives | 2014–2019 |

In 2003, progressively declining hilsa catches led the GoB to adopt the Hilsa Fisheries Management Action Plan (HFMAP), including scientific fish stock assessments and sanctuary areas (where fishing is forbidden during the hilsa breeding seasons), and compensation for fishers adversely affected by these regulations. Also, HFMAP initially banned fishing for 11 years. However, the management plan was implemented with little to minimal input [36]. Then, sanctuaries were established, and hilsa production increased from 0.255 million MT in 2005–2006 to 0.517 million MT in 2017–2018 [37]. Despite this success, many dependent communities remain vulnerable to food insecurity and poverty, particularly during the seasons when fishing is banned [38].

Consequently, this hilsa restoration's sustainability is at risk due to over-exploitation of brood fish during the breeding season; many individuals flouting the fishery ban, non-compliance, and conflicts over resource use [39]. To avoid over-exploitations and to sustain the natural resilience on the hilsa fishery, it is essential not only to understand ecological limits, but to also enhance social resilience, i.e., the ability of individuals and communities to collectively manage their resources, to better cope with disturbances, and to strengthen their means of adapting in the face of socio-economic, political and environmental change [40]. In this regard, the establishment of a functioning co-management (increasing the stake of local actors in decision-making for fisheries management by power-sharing, implying a partnership among fisheries users, related business enterprises, non-government organizations, and

the government regime in the hilsa sanctuaries could enhance their sustainability in socio-ecological terms [41]. Co-management is also widely promoted as a preferred approach for managing complex social-ecological systems associated with small-scale fisheries [42]. Ideally, it combines the best of top-down and bottom-up approaches, linking resource users, government agencies, and other stakeholders through vertical and horizontal connections, and providing mechanisms for collaboration and adaptive creativity [20]. Whether and how co-management could be designed and implemented through power-sharing agreements to mitigate the current sustainability challenges to hilsa fisheries is one of the gaps that the present paper aims to address.

## 3. Theoretical Framework

Different tools and methods (decentralization, power cube, power matrix, peeling the onion, net-map) are used in analyzing power relations among stakeholders in natural resource management [28,43]. Decentralization [43] and Power cube framework [29] are well recognized and used widely [19,27]. Since the decentralization framework focuses mainly on the powers of the government (legislative, executive, and judicial) rather than those of the community, we have selected the power cube framework (Figure 1) as the primary analytical framework for this study. The following section of the present study is modified from the Institute of Development Studies for the analysis of spaces, forms, and places (levels) of power [29].

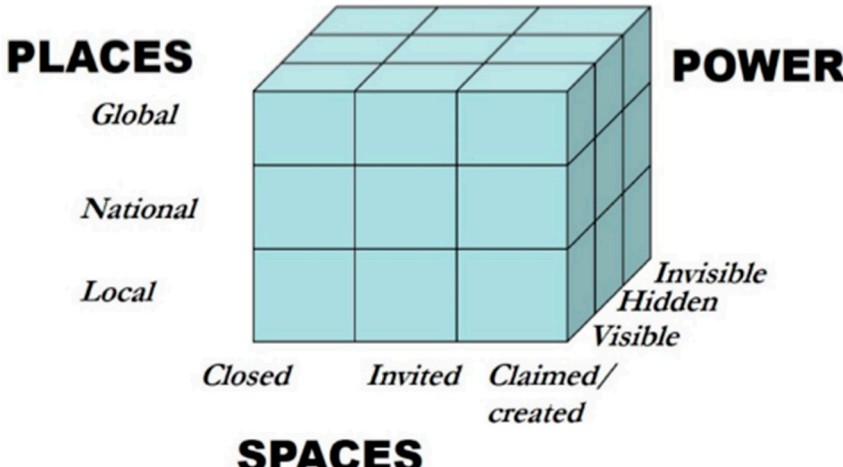

**Figure 1.** Power cube framework [29].

Power cube is a conceptual framework that can be used to understand and analyze how power works in processes of governance and citizen participation in organizations and social relationships [29]. It uses a multi-faceted approach to explore the visible, hidden, and invisible dimensions of power by mapping the various spaces and levels where stakeholders experience and exercise these forms of power. Though visually presented as a cube, it is vital to think of each side of the cube as a dimension or set of relationships, not as a fixed or static set of categories [29] (Figure 1).

In the power cube framework, "levels" refer to the places where participatory action takes place. These may be at the local level, the national level, which involves national networks or national assemblies, and the international level, including regional bodies or systems. This study uses an "adapted version of the Power cube framework," adopting two of the three dimensions of the framework for analysis, namely local level (household, fishing boat crew, village, Upazila referred to as sub-district) and national level (district).

"The 'spaces' dimension, in turn, refers to the potential arenas for participation and action, including what we call closed, invited, and claimed spaces. 'Closed spaces' refer to those decision-making spaces where only the influential stakeholders (politicians, experts, managers, etc.), in closed-door meetings,

can participate in decision-making. 'Invited spaces' are instances where efforts are made to widen participation, move from closed spaces to more 'open' ones, and create new spaces. A wider variety of stakeholders (users, citizens, or beneficiaries) may be invited by various authorities (governments, supranational agencies, or non-governmental organizations). Claimed/created spaces, in turn, are the spaces co-opted by less powerful stakeholders from or against the power holders or created more autonomously a grass-root level [29]."

Finally, the "forms" dimension of the cube refers to how power manifests, including visible, hidden, and invisible ways. "Visible power" includes the definable aspects of political power: formal rules, structures, authorities, institutions, and decision-making procedures. Also, Visible power is understood by the axiom—A has power over B to the extent that can get B to do something that B would not otherwise do. In contrast, "hidden power" involves certain influential people and institutions maintaining their influence by controlling who gets to sit at the decision-making table and what gets put on the discussion and decision list. These dynamics may operate simultaneously at many levels to exclude and devalue the concerns and representation of less powerful groups. "Invisible power" refers to social and cultural norms, perceptions, and beliefs that condition or influence people's individual or collective views.

## 4. Materials and Methods

### 4.1. Study Area

For the present study, fieldwork was conducted in four hilsa fishing communities of Bangladesh, in the villages of Rahmatpur and Sudirpur (Andharmanik sanctuary, Kalapara Upazila of the Patuakhali district—Study Area 1) and Uttar Bagula and Dakxin Bagula (Upper Meghna sanctuary, Haimchar Upazilla of the Chandpur district—Study Area 2) (Figure 2).

The selected communities are directly dependent on fishing inside the sanctuary areas for their livelihood, including fish sales and fisheries-related activities such as fish drying, fish trading, net mending, boat building, and boat repair. With a total population of approximately 12,000 people, Study Area 1 is situated in the south-western part of Bangladesh, along the Andharmanik River, Kalapara Upazila (sub-district), Patuakhali (district) (Figure 2). The Andharmanik River is a famous breeding and nursery ground of hilsa. Hence, it was declared the 4th hilsa sanctuary by the GoB, with seasonal prohibitions on catching juvenile fish (jatka) (November–January) and breeding female hilsa (22 days within October–November, based on the lunar cycle) [44]. Study Area 2, where approximately 8000 people live, is located along the lower Meghna River, Haimchar Upazila, Chandpur district, in Bangladesh's southeastern part (Figure 2). The Meghna River is also considered one of the most significant breeding zones for hilsa fish. The government declared 100 km of the lower Meghna River as a hilsa sanctuary in 2005 to retain juveniles and brood hilsa fish.

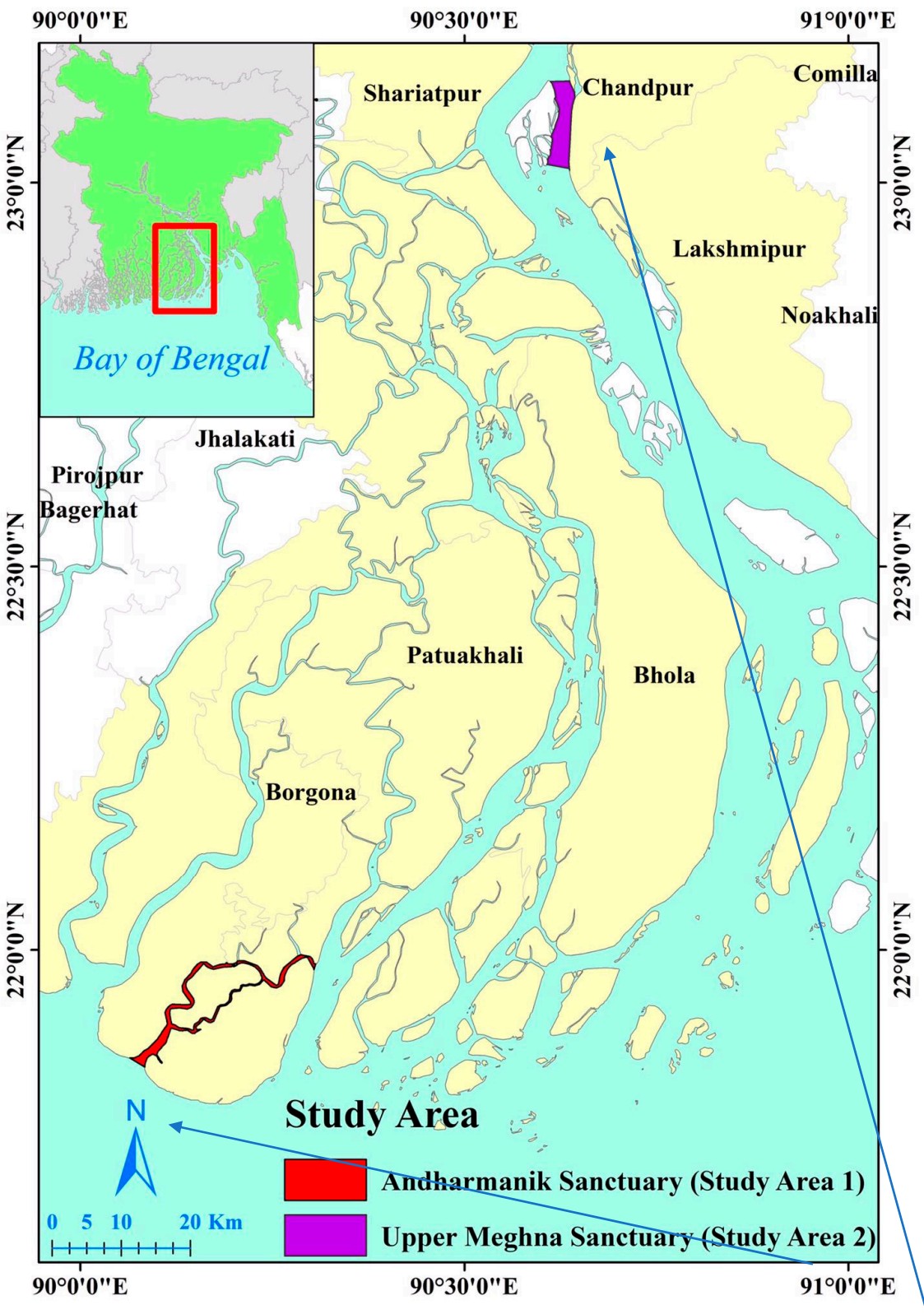

**Figure 2.** Study Area in Bangladesh and a map of hilsa sanctuaries. Adopted from [36] with permission.

*4.2. Methods*

Both primary and secondary data were collected for this research. Data were collected in two stages: in Study Area 1, data were collected from December 2016 to February 2017; in Study Area 2,

from November 2018 to January 2019. For the present study, secondary information on governance (fishery rules and regulations, the establishment of sanctuaries, implementations of different fisheries policies) and power structures (administrative) in the hilsa fishery were collected from newspapers and study reports issued by NGOs and local universities working with the SSFs (mainly the hilsa fishery) in the coastal areas of Bangladesh, and from associated legislation issued by the GoB. Moreover, these data were embedded in a scientific literature review related to power, governance, and historical progression of co-management issues on natural resource management and hilsa fishery management in Bangladesh (see Supplementary Materials, Tables S1 and S2). In this manner, the reviewed literature helped us to better design and contextualize our research questions, interviews, and sampling strategy [45]. It proved to be useful in validating many of our findings [46], which we discuss further below.

Primary data were collected using qualitative methods, including in-depth interviews and focus group discussions (FGDs). "Qualitative methods were used to understand human behavior by analyzing social structures within the framework they take place within their social context [47]. Purposive and snowballing sampling strategies were used to select the interview respondents. A "snowballing" sampling method was used to identify potential fishers to interview because of the diverse group of people engaged in hilsa fisheries [48]. Purposive sampling approach was employed to interview the more knowledgeable fishers [49]. In total, 128 in-depth individual interviews were carried out. Among them, 120 were with hilsa fishery stakeholders. In each of the four villages, interviews were undertaken with 20 individual hilsa fishers—mainly men (n = 15 per village), but also some women (n = 5 per village). An additional ten interviews were conducted with other stakeholders, including fish traders (n = 2 per village), boat owners (n = 2 per village), money lenders (n = 2 per village), local government representatives (n = 2 per village), and local governments administrative personnel (n = 2 per village) (see Table 2). Additional stakeholders were identified based on interviews with fishers. Depending on work location, additional stakeholders were located either in villages or fish landing centers and nearby Upazila government office premises. In conjunction with the hilsa fishery stakeholders, academics (n = 4), local NGO representatives (n = 2), and environmental specialists (n = 2) were interviewed. The interview sample size was determined based on the requirement of information and guided by the principle of data saturation [50].

**Table 2.** Sample distribution of interviewed hilsa fishery stakeholders.

| Stakeholders | Number of Stakeholders in Study Area 1 | | Number of Stakeholders in Study Area 2 | |
|---|---|---|---|---|
| | Rahmatpur (Inhabitants: 7000) | Sudirpur (Inhabitants: 5000) | Uttar Bagula (Inhabitants: 5000) | Dakxin Bagula (Inhabitants: 3000) |
| Hilsa fishers | Men—15 Women—5 | Men—15 Women—5 | Men—15 Women—5 | Men—15 Women—5 |
| Fish traders | 2 | 2 | 2 | 2 |
| Boat owners | 2 | 2 | 2 | 2 |
| Money Lenders/*Dadonder* | 2 | 2 | 2 | 2 |
| Local government representatives (Upazila Chairman, Union Parishad Chairman) | 2 | 2 | 2 | 2 |
| Local government administrative personnel | 2 | 2 | 2 | 2 |

The first author made several visits to the study sites. In the first visit, the first author explicitly stated his fieldwork's objectives and took the respondents' permission for the fieldwork. In the whole interview process, individual interviews were conducted between the respondent and the first author in the presence of one research assistant, which helped maintain the confidentiality of the respondent's information, which also helped build trust with the respondent. Several visits by the first author and his interactions with the respondents in each visit certainly helped build trust and rapport with the communities as rural societies in Bangladesh are open and friendly to their visitors.

Before each interview, respondents were informed about the study and given assurance that its ethical principles and interview data are entirely confidential. The entire interview will be audiotaped/videotaped. Coded numbers will be replaced all names. Any publications and any reports

will not identify the respondents by name. Participants' quotes will be used if found appropriate and related to the study (see Supplementary files—Consent form). Data will not be used for any purpose other than scientific publications. Prior permission was obtained for all recordings of interviews and photographic documentation. Participation in the research was entirely voluntary, and all participants were informed of their right to withdraw from the study at any stage (see Supplementary files—Consent form). Interviews were carried in local dialects and language (Bengali). Each interview lasted approximately one hour on average.

Apart from in-depth interviews, 8 Focus group discussions (FGDs) (2 in each village) were carried out. One set of FGDs was conducted with only hilsa fishers (n = 8), including both men (n = 5) and women (n = 3), and the other FGDs were held with a mix of stakeholders (n = 10) (see Table 3). With the permission of all participants, the FGDs were recorded, and each FGD lasted an hour.

**Table 3.** Sample distribution in the combined stakeholders FGDs (focus group discussions) (n = 10).

| Stakeholder Groups | Number of Participants in FGD |
|---|---|
| Hilsa fishers | 4 |
| Fish traders | 1 |
| Boat owners | 1 |
| Money Lenders/*Dadonder* | 1 |
| Local government representative (Upazila Chairman, Union Parishad Chairman) | 1 |
| Local government administrative personnel (Upazila Fishery Officer /Police) | 1 |
| Local NGOs representatives | 1 |

The pilot interview was conducted with the communities to test the suitability of the questionnaire. The questionnaire was then revised accordingly before administering it. The questions were modified for each interview, depending on the role and representation of the interviewee. Many researchers have reported difficulties in using the word "power" with groups of stakeholders, putting influential stakeholders in the hot seat, and evoking a defensive response [51]. Hence, we attempted our best to communicate to respondents in simple terms what power meant in the context of our study. As an example, we said power is usually understood as the capacity to influence something, or someone and interviewees understood this well. The interviews were semi-structured and open-ended. Also, a list of issues and questions for each interviewee was drawn up concerning (i) distribution of power among hilsa fishery stakeholders (who is the most powerful and who is the least powerful?), (ii) exercise of power among hilsa fishery stakeholders (how do influential stakeholders exercise their power?), (iii) spaces of power(the potential arenas for participation and action by different stakeholders), (iv) forms of power (how is power demonstrated), (v) levels or places of power (where participatory action takes place) and (vi) possible ways to minimize power asymmetries (see Supplementary files). Moreover, the interviews were used to probe on the following questions: (a) current governance systems in the hilsa fishery and loopholes, (b) reasons for a decline in hilsa catch, (c) pros and cons of the contemporary hilsa conservation and management policies, (d) power dynamics bearing on the execution of hilsa incentive-based conservation, (e) money lenders and debt cycle issues of hilsa fishers, and (f) possible measure to increase the long-term sustainability of the hilsa fishery.

For the FGDs, similar questions were developed, but we allowed for new issues to emerge during the discussions (see Supplementary files). The use of FGDs helped ensure that the findings were subjected to debate and dialogue, particularly among those with conflicting views and interpretations of some of the more sensitive issues [52]. During the FGDs, the first author of the present study played the role of a moderator. During the discussions, the moderator ensured that respondents understood questions. Also, the moderator paraphrased or summarized the participants' comments whenever needed. The moderator attempted to remain neutral (as an example agreeing or disagreeing with explanations) to ensure that everyone felt comfortable expressing their opinions. Also, to deal with dominant participants, the moderator acknowledged their opinion and actively solicited others' opinions.

All data collected from the in-depth interviews and FGDs were first transcribed, translated from Bengali to English, and analyzed using thematic analyses for related themes [53]. Thematic analysis is described as a descriptive method that flexibly reduces the data [54]. To support, clarify, and give voice to the respondents' perceptions, we also used direct quotations of the interviewees. Initial analyses of the data were conducted in the field with the respondents to eliminate personal biases in interpretation. Furthermore, such initial analyses were shared in the FGDs to ensure the respondent's statement's authenticity. The data analysis was supported by field notes taken throughout the data collection.

## 5. Results

The presentation of the main findings of our research is structured as follows: first, an overview of the main issues that emerged from the power cube framework exercise (Figure 1): levels of power (in terms of its practice) and distribution, spaces of power, and forms of power. Secondly, the loopholes in the present hilsa fishery governance systems are presented (compliance with regulations).

### 5.1. Exercise of Power among Hilsa Fishery Stakeholders

#### 5.1.1. Value Chain and Power Relations

Fishers, assemblers, processors, traders, intermediaries, transporters, and day laborers are involved in the hilsa fishery. The fishery is capital intensive, most fishers cannot afford to go fishing at their own expense, and fishers work without collateral, meaning they do not have access to standard bank loans. Thus, an informal loan (*dadon*) from a fish trader (*aratdar*) is the only available financing. In return, the *aratdar* can buy the catch at a lower-than-market-value price. Typically, a *mohajan* (owner of a boat and fishing nets) uses this *dadon* system to purchase and maintain his productive assets. Despite this, the *mohajan* often loses money by selling his catch at a lower price and paying a commission to the *aratdar*. Usually, a *mohajan* works as the captain (*majhi*) of his boat or can hire another experienced fisher as *majhi*. Crew members are called *malla* or *vaghi*—general laborers in operation paid either a day wage or with a share of the proceeds from the catch's sale.

During the in-depth interviews in Study Area 1, hilsa fishers stated that the power relations based on wealth and dependency among the actors in the value chain (most powerful to least powerful) are as follows: *Aratdar→Mohajan→Majhi→Malla/Vaghi*. As the aratdar provides loans to the *mohajan*, the *aratdar* can be considered the most powerful in the chain. A similar value chain was also observed in Study Area 2 and confirmed in the FGDs. Furthermore, one elderly respondent in Study Area 1 shared that: "Sometimes conflicts arise between *mohajon* (boat owner) and fishing crew when the latter perceives injustice in profit sharing or wage payment because the former has a connection with a powerful local political leader, thus trying to deprive hired fishers of their compensation".

Although benefits of hilsa fishing are unevenly distributed among different groups along the value chain, *aratdars* as investors are key players in the capital-intensive hilsa fishery in both study areas. The primary limitations of the hilsa fishers are a lack of brokering power and market information at the market level.

#### 5.1.2. Power Relations among Stakeholders

In addition to the main stakeholders in the hilsa fishery value chain described above, there are also other stakeholders involved in hilsa fishery management, local government administration personnel (Upazila Nirbahi Officer-UNO, Upazila Fishery Officer-UFO, Coastguard, and Police), NGOs, local political leaders, local government representatives (members/chairman), transport services, net traders, wholesalers (*paikar*), ice traders, diesel oil merchants, the chief *aratdar* (in Dhaka City), and journalists. All these stakeholders have relations with at least some (not necessarily all) of the other stakeholders, involving multiple power relations (see Figure 3).

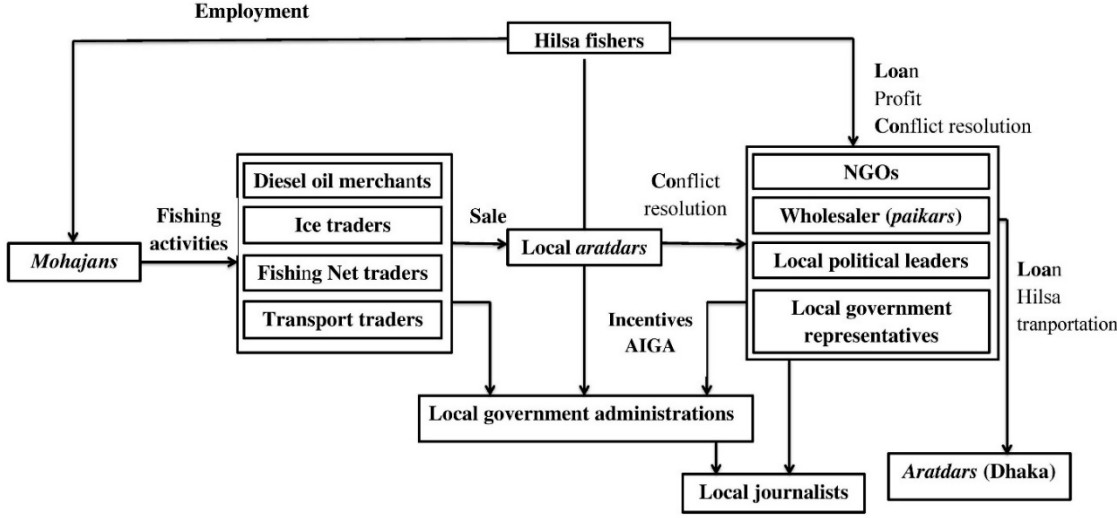

**Figure 3.** Power relations and service provision among stakeholders in the hilsa fishery value chain and management (details are described below).

Based on the in-depth interviews and FGDs in both Study Areas 1 and 2, the relationships and power dynamics are summarized as follows (Figure 3): The *mohajon* has relations with the diesel oil merchants, net traders, ice traders, and transport services to run the fishing boat and preserve the fish for sale to the *aratdar*. In this case, ice sellers, oil merchants, and net traders have more power than the *mohajan*, as the latter cannot run his boat without oil or preserve the hilsa without ice. Secondly, the *aratdar* has relations with the NGOs and wholesalers (*paikars*), as *aratdars* get monetary support from them. *Aratdars* also have ties with the local political leaders and local government representatives to solve conflicts among crew members in the hilsa fishing boat. Hence, local government representatives and local political leaders have power over the *aratdar*. Thirdly, net traders have relations with the local government administration as local government administration enforce the law and seize the illegal fishing nets, such as the monofilament gill nets still commonly being sold. Also, local political leaders have a relationship with local government representatives. Local government representatives depend on local political leaders as local political leader have power to influence local people and fishers during the election period (vote bank). The local government administration is the most powerful as it has judiciary power to implement compliance measures and address management issues. Local political leaders and local government representatives (union parishad members and chairman's) work on the guidelines given by the local government administration personnel.

Besides, wholesalers (*paikar*) have a relationship with the prominent *aratdar* in Dhaka city for more substantial loans, with the local *aratdar* buying hilsa from them, and transport services to transfer the hilsa fish to the *aratdar* in big cities of Bangladesh, including Dhaka. Local journalists have relations with the local stakeholders (political leaders, local government representatives) involved with the hilsa fishery and local government administrative personnel as journalists publish updates of the law and enforcement measures related to hilsa fishery management. Finally, general hilsa fishers (*vaghi*) have relations with the *mohajan* to obtain work in a boat, catch hilsa fish, and to get a profit of the catch; with NGOs to get loans during times of crisis, including the fishing ban period; with local leaders to solve conflicts with their *mohajans*; with local government representatives to resolve social disputes and to be included on beneficiary lists to collect incentive pay for not fishing during the ban period, and; with local government administrative personnel to get incentives and favors for AIGA (alternative income-generating activities). In Study Area 2, a hilsa fisher described position amidst these power relations as follows: "You see, we do not have any power to raise our voice in the fishing boat, the *mohajan* is the boss here, although we worked hard to catch fish in the scorching sun, dark of the night, and stormy weather. We do not have any bargaining power and price of the fish, as the *mohajan* sets it.

We can also say nothing when local government representatives select non-fishers to get compensations provided by the government. We do not have any power to stop the NGO personnel when they take our household assets as we failed to repay the monthly installment of the loan. We are the powerless group in the fishery chain and suffer all year-round. No way to get rid of the misery."

5.1.3. Power Relations among Hilsa Fishery Stakeholders According to Importance and Influence

The level of "importance" and "influence" of stakeholders are not the same for each actor in the hilsa fishery. However, the stakeholders (Table 4) are directly or indirectly engaged with the management of the hilsa fishery. In this study, we refer to those stakeholders from different occupations who are most frequently connected with the fishery by the word "importance". In contrast, the term "influential" applies to those who have positional power (i.e., in most cases, local leaders and administrations are in this category).

**Table 4.** Power relations/power dynamics among hilsa fishery stakeholders according to importance and influence.

| Most important and least influential | Most important and most influential |
|---|---|
| • Fishers | |
| • Boat/net owner | **Most important and most influential** |
| • Crews (*malla* or *vaghi)* | • *Aratdar* (moneylenders) |
| • Local retailers (*paikar*) | • *Mohajan* (boat owners) |
| • Net traders | • Political leaders |
| • Local shopkeeper | • Upazila Fisheries Officer |
| • NGOs | • Local government administration personnel |
| • Respected elderly person in the area | • Local government representatives (Upazila Chairman, Union Parishad Chairman, and members) |
| • Women (hilsa fishers' wife) | |
| **Less important and less influential** | |
| • Fishing equipment's traders | **Less important and most influential** |
| • Transport workers | • The Police |
| • Workers/daily laborers | • Law and Order/Coastguard |
| • Ice trader | • Media (journalist) |
| • School teacher | • Dredging (Bangladesh Inland Water Transport Authority) |
| • Local restaurants owners | |
| • Diesel trader | |

During the combined FGDs in both study areas, the hilsa fishery stakeholders were categorized into four different classes (Table 4) based on their contribution, connectivity, and positional power they exert on the fishery management system. However, there was a contradiction among the participants to plot the stakeholders in different categories. The following final diagram was drawn based on the majority concern and logic. These are: (a) most important and less influential, (b) most important and most influential, (c) less important and less influential, and (d) less important and most influential (Table 4).

Most important and less influential represents those directly connected to hilsa fishery with no/minimal positional power. While the most important and most influential depicts stakeholders with more adherents and power in this system. For example, respondents of hilsa fishers were asked why they rank *mohajan* as one of the most influential? The below answer illustrates the power dynamic: "You know, the *Aratdar* (money lenders) is our local guardian. He gives us a loan to make boats, buy and maintain our fishing equipment, buy daily necessities, and get protective security services whenever needed. Hence, if money lenders asked us not to do something or do something related to fishing or any related matter, we always follow their advice."

Less important and less influential represents those who have indirect contributions and less power as well. Moreover, few stakeholders have more positional power. However, they have less

function/engagement in this fishery system. They are defined as less important and most influential. In the Study Area 2, during the combined FGDs, hilsa fishers stated why they rank Police or Coastguard as less important and most influential: "During the fishing ban period, if we go out for fishing, there are raids, and we are often caught red-handed by law enforcement officials (Police and Coast guard). Moreover, law enforcement officials burn or seize illegal fishing equipment afterward. As they have the power to seize our resources, they are the most influential for us (fishers)."

### 5.1.4. Power Issues between Motorized and Non-Motorized (Manual) Boats

During the FGDs in both study areas, fishers on non-motorized and motorized boats accuse each other of illegal fishing. However, both types of hilsa fishers often continue to fish during the ban periods. Fishers on non-motorized boats can only harvest a smaller catch and are often caught red-handed during raids by law enforcers, due to their limited mobility with a smaller boat. Motorized boat fishers can harvest more fish due to their greater mobility, and they can escape from patrols faster due to the higher speed of their boats. One non-motorized boat owner and a hilsa fisher in Study Area 2 explained these unequal opportunities as follows: "You know, local people with a connection to power usually own all the big, motorized boats. They usually give bribes to the Police and can continue fishing during the ban period, especially at night. If there is any raid, they usually get information from their sources in a police station. We go for illegal fishing during ban period to survive, but they (motorized boat owners) catch fish out of greed, as they have monetary and political power."

### 5.2. Spaces of Power in the Hilsa Fishery

### 5.2.1. Closed Spaces

During our interviews and FGDs, we talked with the participants whether they wanted to participate with the government representative in the decision-making table to sustain the hilsa fishery. We found that stakeholders wanted to participate in decision making. However, it was evident during the in-depth interviews and FGDs that stakeholder participation is mostly lacking. A veteran fisher in Study Area 1 stated: "We know that we are illiterate; however, we gained fishing experience by fishing hilsa in the Andharmanik River for a long period. Local fisheries department officials should discuss hilsa matters with us and use our local knowledge in setting the dates for fishing bans and the geographical boundaries for fish sanctuaries and in devising other fisheries management strategies. But it never happened until now." Another fisher from Study Area 1 added: "Local government administration personnel do not feel any interest in discussing what would be an adequate amount of compensations for us. Usually, we have different numbers of family members. Still, we get an average amount of compensations, even less during the fishing ban period, so that is not enough to meet our necessary expenses".

Such issues were discussed more elaborately during the FGDs in both study areas, and fishers protested about the distribution of compensations: "We know that the government of Bangladesh introduced a compensation program—40 kg of rice per month per hilsa fishing family during the ban period. The compensation rarely comes in time to feed our families during the fishing ban period. Some fishing households have more than 6–8 members. Do you think 40 kg of rice is enough for those households for one month? Sometimes, we get only 30–35 kg of rice instead of 40 kg, if we ask why we are getting only 35 kg of rice? Then authorized persons answered it is due to minimize the official cost/transportation cost or unknown." Also, participants during the FGDs in the Study Area 2 claimed that they urged the local government administration to include only the genuine hilsa fishers in the beneficiary lists for the incentives. However, some deserving fishers are barred from these lists, and some non-fishers are given favoritism, benefits intended as incentives not to fish during the ban.

### 5.2.2. Invited Spaces

Recently, initial efforts were taken by different non-governmental organizations and the GoB to widen the participation of stakeholders in the hilsa fishery value chain in decision-making regarding the sustainability of the hilsa fishery. Firstly, IIED initiatives (2013–2016) with the GoB and other partners helped generate the political will to create sustainable, bottom-up solutions that can preserve hilsa and enhance livelihoods on a regional level [33,35,37,55]. Secondly, to improve the annual incremental production further, the Department of Fisheries (DoF) and WorldFish have jointly been implementing "Enhanced Coastal Fisheries in Bangladesh (ECOFISH-Bangladesh)," a USAID supported project (2014–2019). The project has provided support to the DoF and local communities to establish a science-based "adaptive co-management" approach that focuses on the brood hilsa protection, juvenile conservation, control over illegal fishing gear, and overall ecosystem resilience involving all stakeholders. The initiative also contributed to improving community empowerment focusing on women's access to resources and technologies for livelihood diversification and community resilience improvement to enable compliance during fishing ban periods. One participant in the FGDs in Study Area 1 shared the following: "Including the female members of our family, we need support to compensate for the loss of income during the banned periods. For this purpose, we are encouraged, motivated and consulted by ECOFISH to practice different alternatives livelihood options including net-making, cage culture/fish farming, poultry nurture (ducks and chickens), small dairy projects (keeping goats or cows), starting-up a new business (small grocery shop/tea stall), plant nurseries, gardening, and handicrafts (doll making)".

### 5.2.3. Claimed/Created Spaces

During the FGDs, respondents (mainly hilsa fishers) were asked what they would do to empower themselves, or better to balance the power dynamics in the hilsa fishery. Hilsa fishers in the Study Area 2 responded that: "We must band together during the local government election; we should elect the person as a union parishad member or chairman (local government personnel), honest and fair, and working for the hilsa fishers' wellbeing. The local *aratdar* has immense power. However, if we the fishers can get an opportunity to sell the fish to another *aratdar* for a good price by bargaining, not always to the same *aratdar* that we are bound to, the power of that *aratdar* will diminish. It will make the *aratdar* compromise with the fishers and offer a good price for our fish catch. Also, if we get money as an easy loan from the government, we can buy fishing boats and nets. Consequently, we can gain the power to sell our fish wherever we want to sell".

### *5.3. Forms of Power in the Hilsa Fishery*

### 5.3.1. Visible Power

In this study, we noticed that hilsa fisher (*vaghi*) have relations with the *mohajan* to obtain work in a boat, catch hilsa fish, and to get a profit of the catch. However, it was evident during the interviews that fishers get less profit from the catch as the actual selling price of the catch in the market is not disclosed to them. A *mohajan* has the power, and fishers do not have any bargaining power. Also, fishers do not get the opportunity to sell fish to other *mohajan*. One elderly fisher in Study Area 1mentioned: "We worked hard in the boat; even we are bound to work during the bad weather. However, when it comes to the payment option, we always deprived. If we shout, *mohajan* replied with a loud voice that he would not provide any monetary help as a loan during our crisis time. *Mohajan* has political power as he has a relation with the local leaders, and he has good contact with local administration. Although we want our catch's good price by bargaining, we cannot compete with *mohajan* for his such visible power."

### 5.3.2. Hidden Power

We found that local *aratdars* have exercised hidden power in hilsa fishery management. During an in-depth interview with an *aratdar* in Study Area 2, he said that he supports the seasonal bans on fishing for hilsa (juveniles and broodstock) and prohibitions on the use of jal nets imposed by the government. Yet a hilsa fisher in that same area stated in an interview that: "*Aratdar*, you know, they are the boss in our locality. We take a loan from them for different purposes. When it comes to the ban period, we request them to decrease the amount of installment or extend the due date for a few months. However, they never listen to or consider our request at all. Moreover, they force us to go fishing during the ban periods, using illegal nets. They have a sort of hidden power that we can't see."

### 5.3.3. Invisible Power

We also found that there is a sort of invisible power among women fishers. They were more open than the male informants to social change and finding alternative means of family subsistence. They were more ready to start the implementation of alternative income-generating activities (AIGAs). The focus group discussions in Study Area 1 revealed that the more women became involved in fishery management matters, the more AIGAs were implemented, increasing overall household wellbeing. These activities include net-making, cage aquaculture, poultry rearing, small dairy ventures, plant nurseries, gardening, and handicrafts. These fisherwomen felt that they could play a more substantial indirect role in conserving hilsa by supplementing their family income by other means.

### 5.4. Governance and Compliance with Regulations in Hilsa Fishery

### 5.4.1. Loopholes in Hilsa Governance

Recently, hilsa production has increased, and according to participants' perceptions, it is evident that government initiatives since 1950 (The Protection and Conservation of Fish Act—PCFA) to 2019 (Enhanced Coastal Fisheries in Bangladesh—ECOFISH-BD project) have played a significant role. However, during the in-depth interviews and FGDs, participants shared their opinions about the hilsa fishery governance regime's loopholes. Participants in Study Area 1 summarized the reasons for non-compliance with fisheries regulations as being mainly due to poverty among fishers; owing money and patron-client relationships with intermediaries—mainly *mohajan*; irregularities in incentive distribution and limited opportunity of alternative occupations during the fishing ban period; availability of illegal, destructive fishing gear and greed for larger catches, especially concerning broodstock; corruption amongst some members of law enforcement agencies; hilsa fishers being excluded from decision-making processes. One academic hilsa researcher in Study Area 2 raised an important point that deserves further consideration related to sanctions and compliance.

"In my opinion, the penalties incorporated into legislation are insufficient, and the financial penalties are outdated. Also, the power and functions of the fishery officers and staff in the implementation of legislation and regulations are usually not specified by the respective laws and regulations. As a result, the proper implementation of fishery laws and policies is not fully ensured. Hence, the implementation should be extended to the district fishery officers (DFOs)."

Similarly, in another in-depth interview conducted in Study Area 1, one local government administrative personnel who was directly involved with the hilsa fishery management expressed his opinion about the loopholes in existing fishery governance as being the following:

"People think that we have huge power. However, we face numerous problems due to not imposing appropriate penalties for breaching the rules. The punishments stipulated in the existing laws are not always properly applied by law enforcement agencies if bribes are taken from the fishers not to reduce the penalties. We also go patrolling during the fishing ban period, but sometimes sudden patrol campaigns have been unsuccessful due to leaking of information about them."

Moreover, a local NGO representative in the Study Area 1 stated that the respective officials rarely inspect fishers' boats and penalize the fishers for a breach of regulations. In his opinion, such weak

enforcement has led to fishers violating the law with impunity, as fishers realize that the chances of being caught and punished are very slim.

### 5.4.2. Gaps in the Execution of Hilsa Incentive-Based Conservation

To protect the hilsa fishery from the overexploitation of juveniles and brood, the GoB established six hilsa sanctuaries (first four sanctuaries in the year 2005, fifth and sixth in the year 2011 and 2018 respectively) in the Padma, Meghna, Tetulia, Gazaria and Andharmanik rivers. In the sanctuaries, a fishing ban on catching jatka (juvenile hilsa with a size of <25 cm) has been imposed on the sanctuaries from 1 November to 30 June each year. Another 22-day countrywide prohibition on fishing for brood hilsa is implemented each year during the lunar month beginning in October to ensure safe breeding for hilsa. Each year, law enforcement agencies run a few mobile courts and other operations to seize illegal jatka or hilsa catches and file cases against offending fishers under the Mobile Court Ordinance (2007). Many of these fishers are given imprisonment, fines, or both. Concerning this, a respondent in Study Area 1 expressed his views about the ban periods as follows: "We are aware of the power of the local government administrative personnel. However, the ban period is one of the biggest shocks in our life, and it has placed us under severe economic pressure. To overcome this, we sell off property belonging our to families, take part in seasonal migrations, lessen our daily food intake, force other family members to get jobs, take out loans from money lenders and NGOs at very high interest rates, and, finally, we go fishing illegally, using illegal nets, including for juveniles and broodstock during the ban period, even though we know how powerful the local government administration personnel are and the punishment they can give."

To compensate for hilsa fishers' losses during the fishing bans and to encourage them to comply with the regulations, the GoB provides economic incentives declared as compensation schemes to the affected fisher households, in the form of food (40 kg of rice per month per hilsa fishing family) during the jatka ban period. The GoB also provides start-up money for AIGAs for activities such as net making, raising poultry, raising goats and cows, machine sewing, plant nurseries, kitchen gardening, and cage aquaculture. Understanding the power dynamics in economic incentives distribution for the hilsa fishery management also requires understanding the government institutions involved in, or play critical roles in, incentive distribution.

DoF Bangladesh is the lead agency for incentive distribution. However, various other government agencies support its implementation, such as the district administration, sub-district administration, district and sub-district level disaster management, and local union parishad (the lowest tier of the administrative hierarchy). The local fishing community leaders extend help to provide incentives to fishers during the fishing ban period, in the form of rice and alternative income-generating support. The country's Navy, Coast guard, River Police, Rapid Action Battalion (RAB), Air force, and Border Guard Bangladesh help run mobile courts to enforce the fisheries regulations. DoF implements the project through its three units: the central office (based in Dhaka), the district fisheries office, and the Upazila fisheries office. However, in the FGDs in Study Area 1, hilsa fishers complained about the distribution of incentives: "The compensations do not come in time to support our families during the fishing ban period. More often, we get less than the amount of rice that was allocated to us. Local government administration, Union Parishad, determines who is on the beneficiary list for the incentives program. There are favoritism and corruption in the preparation of that list. Because of a connection with political power, a section of non-fishers is included in the beneficiary list. At the same time, many marginal fishers are left out. Hence, such irregularities in the compensation scheme create social tension."

### 5.4.3. Ambiguities in Compliance with Regulations

There are several protection and conservation measures for hilsa based on the Protection and Conservation of Fish Act (1950) and the Marine Fisheries Ordinance (1983). These include closing some areas to fishing, restrictions on fishing gear, limits on the fishing season, and regulations for

fishing vessels. Regarding the restrictions on fishing gear, the use of a current jal (a fishing gillnet made of monofilament synthetic nylon fiber) with a mesh size of less than 4.5 cm was banned in 1988. The 2002 amendment to the 1950 Fish Act states that "no person shall manufacture, fabricate, import, market, store, carry, transport, own, possess or use a current net." The punishment for violating this law is imprisonment and a fine. A government empowered fishery officer has the same power of investigation, search, and seizure concerning this sort of offense as a police officer with the rank of Sub-Inspector. However, fishers stated during the FGDs in Study Area 2: "Even though current nets are banned for all since they are the main reasons for hilsa stock depletion, law enforcement personnel do not seize these nets from the market, only from us. If law enforcement personnel take steps to seize all the illegal nets from the market, no one can buy it from the market. We think the punishment should be increased for the manufacturing, fabrication, import, marketing, storing, carrying, transport, owning, possession, or use of a current net."

## 6. Discussion

### 6.1. Good Governance

Our findings revealed that the reason for non-compliance with fishery regulations is poverty among hilsa fishers. Since they cannot support their families while abstaining from fishing during ban periods to protect brood hilsa or jatka, they take high-interest loans from an *aratdar*. This *aratdar*, along with their regular *mohajan*, forced fishers to break the rules and fish illegally to repay the loans. The *aratdar* and *mohajan* are themselves, power holders, with secure political connections, and they can thus manipulate law enforcement and policy implementation through bribes and other forms of corruption. In fishing households that try to abide by the rules, government compensation for wage loss often arrives late, and corrupt officials tend to skim off part of the amount that has been theoretically allocated to them.

Despite several problems with the hilsa fishery governance, the present study—using FGDs and in-depth interviews—has attempted to identify some possible means of improving the situation and realizing the goals of good governance initiatives in the hilsa fishery. Participants urged the complete ban of monofilament gill net production in local industries. They also called for an official end to widespread and tacit corruption, especially among law enforcement and support distribution authorities. Participants were urged to employ Coastguard instead of Police for law enforcement. Also, suggestions were made for laws to be established, including punishment for the local government representative if they fail to distribute the incentives equitably. Participants voiced that proper and fair compensation (counting all the family members in the family and the average amount of rice consumed in each household) support needs to be paid to those unable to work during periods when their livelihood is banned. Moreover, credit needs to be made more readily available to buy fishing boats and gear and enhance alternative livelihood opportunities.

Considering the above findings, and based on the existing literature in fisheries governance, we suggest the following initiatives for sustainable and effective hilsa fishery governance in Bangladesh:

Legal framework—Traditionally, in fishery management, legal frameworks are enacted to control activities, reduce conflict, for instance, and control harmful events in the environment. These include gear regulations, such as to prevent the use of nets with small mesh and certain types of trawls, restrictions to fishing vessels, and closed fishing seasons, which are commonly used to enable fish stocks to recover each year [56]. Most of the legislation and regulations governing coastal and marine resources, including the hilsa fishery, were enacted a long time ago, and have been amended to keep pace with changing circumstances. In the context of rapid changes in social, economic, cultural, and environmental conditions, the legal framework now appears insufficient for tackling new challenges [57]. There is an urgent need for an updated legal framework for hilsa fishery management to ensure all stakeholders' well-balanced participation with the fishers' direct involvement.

Better governance mechanisms—If the traditional governance system remains unchanged, arrangements can still be created to improve governance's overall quality [58]. Participation, accountability, coherence, and effectiveness have been recognized as the hallmarks of good fisheries governance [59]. Thus, mechanisms to enhance these features in the hilsa fishery governance would need to be introduced. When all the community groups can voice their interests, such participation ensures a higher degree of legitimacy and compliance with rules that are negotiated [60]. Furthermore, greater participation is essential if hard decisions are to be taken, to make the decision-making process more open, less hierarchical, and more decentralized [58].

Introduction of EAFM—An Ecosystem Approach to Fisheries Management (EAFM) links fisheries management across jurisdictions and boundaries, helping to gain political and stakeholders' buy-in to fisheries [61]. It further increases support for better governance that can lead to better compliance and enforcement [62]. EAFM reduces conflicts, helps unlock financial resources for fisheries through proper planning, and this momentum, in turn, gives rise to support from governments, donors, and NGOs [63]. The Bangladeshi government can take initiatives to implement an EAFM with the help of international donors and NGOs to establish a balance between ecological-social wellbeing and good governance in the hilsa fishery.

Transboundary initiatives—Hilsa is a transboundary fish as it migrates through the rivers of Bangladesh, India, and Myanmar, and is harvested in the Bay of Bengal by all three countries [64]. Millions of fisherfolk are engaged in hilsa fishing in these countries. Therefore, a transboundary initiative for developing a common management policy for hilsa fisheries should be promoted. Such transboundary efforts would help build better governance and a non-exploitative, balanced power dynamics in all three countries for the wellbeing of fishers and ecosystems. Moreover, joint fishing bans should be coordinated and implemented at the same time in all three countries.

## 6.2. Co-Management for Coping with Power Asymmetry

Power-sharing is recognized as a critical aspect of co-management [65]. It is widely accepted in the literature that co-management is about "power-sharing" between local communities and government, and that co-management can be built upon varying degrees of power-sharing [66–69]. Co-management has been widely proposed when other management approaches have failed or have been found less useful in addressing management challenges [70]. Although co-management has benefits, it is not a panacea for fisheries management. Co-management has mixed outcomes, as its context, role, efficacy, and success vary widely [71]. For example, such practices may have some undesirable social and ecological outcomes such as the risk of elite capture and dominance by the powerful, creating incentives for over-exploitation that may increase social inequality and create other conflicts [72]. Also, sharing power and responsibility are essential characteristics of co-management, with enabling legislation, participation, representation, and empowerment is seen as being critical for success [73]. Based on the case study here, we propose that the following be considered concerning the equalization of power relations among the stakeholders, to better initiate and implement co-management arrangements in hilsa fishery governance:

Power-sharing can be arranged at different levels according to the extent of the resource-using community's participation in decision-making processes [67]. This can range from the lowest level of "informing", where the resource users are passive actors in co-management and are informed about what government has decided to do, to the highest level of partnership, where genuine political power is delegated to resource users [74]. Equal sharing of power only occurs when the resource users have the same rights as the government in making decisions on resource management [74].

Fisheries management involves balancing the competing demands of different users of fishery resources. Conflicts among fisheries stakeholders arise due to differences in power, interests, values, priorities, and manner of resource exploitation [75]. Conflicts also emanate from institutional failures in managing fisheries and enforcing laws and regulations [69]. The lack of conflict management mechanisms in the hilsa fishery case relates to problems that reduce the sanctuaries' effectiveness, as

there is rampant non-compliance by users. It also compromises the legitimacy of the conservation measures, as powerful actors such as *aratdars* are often involved in illegal fishing but do not face any penalties, which makes general fishers resentful towards conservation measures.

Most of the hilsa fishers lack the means to operate in their profession without outside capital. Thus, to continue fishing, they must seek credit either from NGOs or the local *aratdars* with their restrictive terms and high-interest rates. From the present study, it was evident that the local *aratdar* has immense power. However, if the fishers can get the opportunity to sell their fish to another *aratdar* at a better price by bargaining, rather than the same *aratdar* they are bound to by a credit contract, the power of the *aratdar* will diminish, and he is more likely to compromise with the fishers and offer a better price for their fish.

Most of our respondents consider their income from fishing as limited. Fishers must be provided with adequate compensation for their financial losses, enabling them to meet their basic subsistence needs during periods when fishing must be temporarily banned. It is evident from the present study that hilsa fishers catch fish illegally or use destructive fishing gear not because they want to, but because they feel it is the only way they can survive. More robust schemes to enable local fishers to lift themselves out of poverty are needed.

Fishers' traditional knowledge, experience, observations, and opinions should be integrated into fishery management policies and implementing those policies. For example, such knowledge may help set the dates for hilsa fishing ban periods and the geographical boundaries for fish sanctuaries and formulating other fishery policies.

*6.3. Limitations of the Study*

The analysis presented in this paper has several limitations. The use of qualitative methods enabled the collation of in-depth perspectives on the values held by the respondents. Many of these respondents attempted to justify the power asymmetries in their societies, their pursuit of a more significant share of local decision-making powers, and their illegal and unsustainable fishing practices. How broadly these attitudes are shared in a wider population is difficult because of the relatively small sample size involved in this study. Further research would thus be needed to subject our findings to a more robust quantitative methodology.

## 7. Conclusions

The power cube framework reflects on levels of power (local, national, global), spaces of power (closed, invited, claimed/created), and forms of power (visible, hidden, invisible). While the framework seems rather straightforward conceptually, its empirical application is complex. In the present paper, we attempted to operationalize the power cube framework, leading us to some theoretically relevant and interlinked conclusions. First, the power cube application revealed subtle power relations that are not self-evident and that occur between a wide variety of actors. In this study, various actors in hilsa fishery were approached to pinpoint crucial issues in the actors' power relations. Secondly, this analysis demonstrated that power relations in the hilsa fishery are highly asymmetrical and unequal, with the least power held by the fishers. Thirdly, these findings led us to seek answers to why local fishers remain without power, which we explain below.

In terms of their economic, social, and cultural value, the hilsa fisheries constitute one of the most, (if not the most), critical coastal and marine resources in Bangladesh today. This paper uses the hilsa shad fishery as a case study to better understand the governance and power dynamics in a small-scale fishery and formulate a policy for sustainable management of this and similar coastal fisheries. Our findings demonstrate that, at present, the hilsa governance structure is far from balanced, with some actors exercising more power over others by closing spaces or creating spaces, sidelining general fishers, encouraging non-compliance and illegal activities, and ultimately doing harm to both the fishery and its users. What hilsa fishery governance should rather be doing is creating a win-win situation for all stakeholders involved in this industry, including the fishers, fish traders,

equipment and supply merchants and financiers in the private sector, and government administrators and law enforcement officials in the public sector. Toward this end, through adaptive co-management arrangements, the fishing community, various stakeholders in the value chain, and the government agencies should all be brought on board to make a coordinated effort to build the sort of hilsa fishery governance structure that keeps the community resilient and the fish stocks sustainable.

For effective co-management, it is necessary for the government at both national and local levels to be aware of the limitations in institutional design and the legislative frameworks that limit the power of the communities they are supposed to serve. From there, they need to bring about the changes necessary to facilitate genuine co-management. In practice, hilsa fishers are also unable to exercise any share of powers due to a lack of legal support and the influence of other interests at higher levels of government. Hence, the present study suggests that policies need to be changed at the national level to fully support a co-management system and enable fishers to exercise powers and raise their voices in decision-making. An effort to develop a gender-inclusive management structure, where both women and men can contribute to decision-making, may further accelerate co-management in the SSF. In the present study, we specifically argue that redistribution of power necessary for the sustainability of the hilsa fisheries and establishment of co-management is the potential strategy for this. There may be similar studies in other parts of the world. Still, we believe this is the novelty of our research in the Bangladesh context.

Although this study focuses on Bangladesh's four coastal fishing villages, the results are potentially applicable across a broader perspective with a similar tropical context. While the case we examined focused on the hilsa fishery in Bangladesh, we believe that the operationalization of the power cube, as demonstrated in the present paper, can also be applied in other studies to reveal subtle ways in which asymmetrical power relations are manifested in the realities of local communities and actors which consume and are dependent on the same natural resources.

**Supplementary Materials:** The following are available online at http://www.mdpi.com/2071-1050/12/14/5738/s1, Table S1: The scientific literature related to the power dynamics, governance and co-management, Table S2: The scientific literature related to the hilsa fishery management in Bangladesh from 2001–2020.

**Author Contributions:** This article is based on the first author's Ph.D. studies. M.M.H.M. designed the research, developed the questionnaire, collected data, analyzed data, compiled the draft, and finally revised and checked the manuscript. A.P. supervised, read, and edited the manuscript. M.A.W. supervised, read, and edited the manuscript. P.S. supervised, read, and revised the manuscript. Additionally, S.S. and M.M.I. read and revised the manuscript. All authors have read and agreed to the published version of the manuscript.

**Funding:** To carry out the fieldwork, the first author received travel funding from the Doctoral School in Environmental, Food and Biological Sciences (YEB) and Doctoral Program in Interdisciplinary Environmental Science (DENVI), University of Helsinki, Finland.

**Acknowledgments:** We sincerely thank all the interviewed hilsa fishery stakeholders. We acknowledge USAID-funded Enhanced Coastal Fisheries in Bangladesh (ECOFISH-Bangladesh), an activity jointly implemented by WorldFish and Department of Fisheries (DoF), Ministry of Fisheries and Livestock, Bangladesh for the logistical help they provided during data collection. We are grateful to Abu Sadek, Alam Pervez, Mohammad Motahar Hossain, Arifuzzaman and Mohammad Saifur Rahman), Faculty of Marine Science and Fisheries, University of Chittagong, Bangladesh for their help during the field work and data collection. We also appreciate the guidance during the field work provided by Mohammad Muslem Uddin, University of Chittagong, and Md. Nahiduzzaman and A.B.M. Mahfuzul Haque, WorldFish, Bangladesh. Thanks to M. Belal Hossain and As-Ad Ujjaman, Noakhali Science and Technology University, Bangladesh for their advice to draw the present study's figures. MMI would like to acknowledge the Too Big to Ignore Global Partnership for Small-Scale Fisheries Research (TBTI). We would like to thank the anonymous reviewers for their thoughtful comments and efforts towards improving our manuscript.

**Conflicts of Interest:** The authors declare no conflict of interest. The founding sponsors had no role in the design of the study; in the collection, analyses, or interpretation of data; in the writing of the manuscript and in the decision to publish the results.

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
