# Peer review of "Governance and Power Dynamics in a Small-Scale Hilsa Shad (Tenualosa ilisha) Fishery: A Case Study from Bangladesh"

_sustainability, doi:10.3390/su12145738_

Round 1

Reviewer 1 Report

The current study conducted a number of interviews with key hilsa fisheries stakeholders at 4 sites within Bangladesh to highlight current systemic constraints in power dynamics along the hilsa supply chain and puts forward suggestion for more effective management and an equitable governance framework.  

The study is interesting and highlight the numerous challenges faced in balancing the basic needs of the most vulnerable communities and those of conservation. Notably it is explicit about corruption and its undermining influence from the perspective of ecological and human dimensions. This is a topic that is often avoided and the authors are to be commended for being clear about its role and importance. The study largely builds on extensive studies in the country on the same topic. In this regard, I have a number of concerns. Notably the authors should clearly reference, in the main of the manuscript, the study by IIED and how the current work expands on this initiative. It would have been interesting to know to what extent recommendations made in publications arising from IIED's work are corroborated by the authors' findings. Perhaps most importantly, it is critical for publication consideration that the approval by the relevant ethics board of human research be included in the acknowledgments. Similarly, it is necessary for the authors to clearly indicate that those individuals that are quoted in the study did consent to those quotes being used - especially in instances where individuals can quite easily be identified.

I suggest the manuscript should undergo major revisions before being considered for publication - and should only be published if the above items are adequately addressed.

Further comments and suggestions are included in the appended documents and below and should be clearly addressed in the authors' revisions.  

Ensure to include topic sentences - these are essentially very succinct summaries of what the next paragraph entails. Line 67 is a good example of a topic sentence.

Line 48 - the first sentence is referring to a specific threat. The topic sentence here should be taking a step back and could be something along the lines of : SSF face a number of threats. Then go into the specific threats

Line 52 and ff - "lack of financial 52 sustainability, inadequate equipment and infrastructure, and limited access to markets" - are these all really new pressures? or what do you define as new?

Line 54 - it is not quite clear why from the arguments presented at least, fisheries should be managed in ethical ways. In other words social disparities and other relevant human dimensions within SSF have not yet been brought to the fore.

Line 58 - it may be worthwhile to include your definition, especially in the context of this study, of governance and how you distinguish it from, or how it relates to, management.

Line 77 - The SSF in question has not yet been introduced

Line 89 - what is meant by spaces? Suggest rephrasing for improved clarity

Line 97 - Given the types of questions raised in the questionnaire, there appear to be a number of issues with the fisheries that go much beyond what is summarised here.

Line 99 - https://pubs.iied.org/pdfs/16622IIED.pdf states that the fishing industry and processes account for 4.3%

Line 99 - hilsa fisheries also account for 11% of total fish catch! Please include some background as to trends in the hilsa fishery and point to the fact that the HFMAP initially banned fishing for 11 years. Most readers are not familiar with most aspect of this fishery so the summary needs to include key points

Line 105 - one of the key missing pieces of relevance to the context of your study is that this management plan was implemented with little minimal input

Line 106 - I would contend that fisheries stock assessments in their own right do not represent a conservation measure.

Line 109 - this sentence comes as a little of a surprise. It needs a better preamble - is the fishery still considered overexploited despite existing measures? ... and is there a history of non-compliance given the vulnerability of some communities? Provide a clearer link between this sentence and the preceding one.

Line 111 (and line 118) - similarly here reference is made to 'further collapses', yet no indication has been given to collapse. Further down in the text sustainability challenges are mentioned. It may be worthwhile to provide some grater context as to historical trends, pointing to previous collapses in the specific fishery of interest or elsewhere and providing some greater context as to what the sustainability challenges entail.

Line 115 - the assumption is made a priori that co-management could enhance the sustainability of the fishery - while pointing to relevant references, I would suggest to include a short sentence indicating what co-management is and that previous work has pointed to co-management as being a useful fisheries management approach to support sustainability as it  recognizes the participation of fishers, local stewardship, and shared decision-making.

Line 130 & line 134 - please reference http://www.powercube.net/wp-content/uploads/2010/01/PowerPack_web_version.pdf at the end of your sentence

Line 142 - much of the text that follows is taken directly or paraphrased from http://www.powercube.net/wp-content/uploads/2010/01/PowerPack_web_version.pdf, it therefore needs to be explicitly referenced. I would suggest a statement up front that indicates that the section that follows is derived or modified from said reference

Line 144 - I would suggest to keep away from value statements such as 'most elite'

Line 157 - what list? decision making list? would suggest to be more specific

Figure 1 and 2 - do the authors have the rights to reproduce the figures if they are taken from elsewhere? The resolution of figure 2 is really low and the sanctuaries not clear. I would highly recommend for the authors to reconstruct these maps and omit all the colours except where necessary e.g., sanctuaries, so these stand out more and to include a legend in the figure.

Line 170 - it is not clear which one is which - i.e., which study area is the protected area & which the space for the conservation and management for hilsa fisheries resources, or do both sanctuaries do both? If the latter then this needs to be expressed more clearly.

Line 173 to 183 - I suggest this come earlier, right after line 166, and for the text lines 169-173 to come after this text. It is more intuitive and logical this way

Line 190 - 'experienced the event in question' - this does not make sense. What event is being referred to? Qualitative methods typically seek to understand human behaviour by analysing social structures within the framework they take place, within their social context (Flick et al. 2004).

Flick U, Von Kardorff E, Steinke I. A companion to qualitative research. Thousand Oaks (CA): Sage Publications Ltd; 2004.

Line 195 - how were the number of stakeholders to interview by stakeholder category arrived at? Please clarify. Given that the study focused on two villages - how many people live in these villages?

Table 1 - please include total number of inhabitants per village (or ideally estimated number of fishers).

Line 200 - how were these additional stakeholders identified? Did they also reside in the villages? please include. Also please clarify if the interview process was developed in partnership with the village communities, or independently.

Line 203 - please cite relevant human research ethics approval from your university in conjunction with the interview process in the acknowledgments including the no. associated with the approval letter from your university. I cannot support the publication of this piece of research without clear evidence of such an application having been made and approved.

What are the authors' relationships to the fishery / stakeholders in the region? How was trust established to ensure that questions were answered in a truthful way? From what I can tell, Md Abdul Wahab was also part of a project with IIED. How does the current research align with that project? Please clarify and include a brief segment concerning IIED's initiative as it is closely related / tied to the work presented here. In this context, the following papers should be cited in the main text (rather than just in the Supp Info tables):

  • Mitigating unintended local economic impacts of the compensation scheme for hilsa management, Essam Yassin Mohammed, Chowdhury Saleh Ahmed, Md Liaquat Ali (2015), IIED Briefing
  • Economic incentives for sustainable hilsa fish management in Bangladesh: an analysis of the legal and institutional framework, Monirul Islam, Essam Yassin Mohammed, Liaquat Ali (2014), Working paper
    Direct economic incentives for sustainable fisheries management: the case of Hilsa conservation in Bangladesh, Essam Yassin Mohammed, Md. Abdul Wahab (2013), Issue paper
  • Power, profits and payments for ecosystem services in Hilsa fisheries in Bangladesh: a value chain analysis, Ina Porras, Essam Yassin Mohammed, Liaquat Ali, Md. Shahajat Alib, Md. Belayet Hossain (2017) Marine Policy
  • Leave no one behind: power and profits in hilsa fishery in Bangladesh: a value chain analysis, Ina Porras, Essam Y Mohammed, Liaquat Ali, Md. Shahajat Ali, Md. Belayet Hossain (2016), IIED Working paper

Line 217 - How was the understanding of power as outlined in this paper and its different forms communicated? It is a fairly academic and structured approach that may not be intuitive to many. Were some standardised simple definitions and explanations provided upfront? if so, expand on the approach and include. Also, please clarify what approaches were used to ensure interviewees understood the framework / concepts they were being asked about. It is one thing for an explanation to be provided, another to be able to apply a concept to be applied to one's environment and context. Her as well, please clarify.

Line 219 - I have some reservations about the questionnaires, in terms of questions asked as well as structure. Please refer to the document itself for queries. The questions themselves are poorly phrased and often unclear. The document makes me question whether human research ethics approval was indeed sought and obtained for this work.

Line 228 - I do not see this in the Supp. Info file, in fact the word trust is not mentioned once, only power (17 times). Also the questions seemed pretty direct to me (e.g., yes/no answers), so please clarify to what extent and how you consider your questions not being direct.

Line 232 - how were these discussions facilitated? please include.

Line 235 - data or information? Please clarify and please clarify specifically what information related to power and governance was extracted

Line 235-241 - this is important and needs to be mentioned further up - I would suggest to place this secondary piece first, since it informed your data collection on site.

Line 249 - all of them? This is unlikely so please indicate how this was operationalised

Line 270 - so the value chain is based solely on the response provided by 1 fisher? Please clarify.

Fig 3 - I am not clear why or how this figure represents power relations (apart from at the center) rather than just existing connections among stakeholders in the hilsa fishery value chain overall. Please rephrase to clarify this clearly. Also include in the legend how the value chain was arrived at and whether it is representative of both study areas (or not).

Line 310 - this sentence is not clear, please rephrase

Line 312 - the only arrow to journalist is with local government administration - yet in the text it mentions connections with other stakeholders. Please clarify

Line 344 - please clarify

Line 362 - the IIED deserves mention in this section given its remit and objectives

Line 404 - this is not really a role per se, it's rather the way in which they influence fisheries management. Please rephrase

Line 415 - Please spell out AIGA

Line 430 - the numbering is not consistent

Line 443 - the local UFO - what does UFO stand for. Also, by citing this statement it means the individual can be identified. Has this person agreed to be quoted? In fact have any of the participants agreed to be quoted in this work?

Line 517 - ?? compelled? Please rephrase

Line 527 - did they have any suggestions as to how this should be undertaken?

Line 535 - not necessarily - they are also enacted to simply control activities and reduce conflict for instance

Line 659 - while this study has highlighted this, my sense is that multiple studies on hilsa fisheries have done this previously. What is different about this particular study - and what are your specific recommendations for tackling this moving forward       

Reviewer 2 Report

Manuscript is verbose while contents are very common in the small scale fisheries in developing countries. Should simplify with scientifically importance. I have major doubts as follows.

Authors emphasize that this study has taken “Power cube framework”, but as authors has written in Line 139-141, authors have given up to consider one dimension of the cube. Therefore this is no longer “Power cube framework” study but “Two-dimensional framework” study. Should revise.

One-sided opinions(complains) from purposely sampled fishers (snowball sampling) are presented. I and readers are not able to judge if these opinions are fair or not because we can not know ideas/responses of related stakeholders to these opinions. Quantitative description of their economic condition is also desirable to understand their opinion.

For instance, “Local government administration personnel do not feel any interest in discussing what would be an adequate amount of incentive payments for us. Usually, we have different numbers of family members. Still, we get an average amount of incentives, even less during the fishing ban period, so that is not enough to meet our basic expenses.” , this just sounds his/her complains if these is no quantitative facts. Should present what is the adequate incentive per family members when authors consider the social condition in the community.

It is difficult to understand the whole picture of the fisheries and community because this is qualitative research. Readers need to know the number of people in each stakeholder category because number is frequently a source of the power.

I believe Figure 3 does not properly present the “Power relations” in the stakeholders. Because I do not believe no any relations between “Local political leader”, “Local government representative” and “Local government administration”. In addition, “Prominent aratdar in Dhaka city” is in this figure although authors have stated that “level” is limited at the local.

Round 2

Reviewer 1 Report

Please see attached document with my responses in blue.

Some clarifications are still needed (minor revisions) before this article can be published.

It would be helpful to get clarity on the difference between governance and management as perceived by the authors. While a definition of governance is provided - throughout the manuscript governance and management seem at times to be used interchangeably. This is confusing.

I did not see the consent form in the list of provided supplementary files.

Reviewer 2 Report

Figure 3 is revised but still very bad quality and no readers would understand from this. Authors should improve this figure much more by referring person correlation diagrams published in other journals.

Author Response

Response to Reviewer 2

We thank the Reviewer for taking the time to read our article and provide some helpful and valuable comments, which we believe have helped to improve the article further. Below we provide our responses to the comments:

Point 1: Figure 3 is revised but still very bad quality and no readers would understand from this. Authors should improve this figure much more by referring person correlation diagrams published in other journals.

Response 1: We thank the Reviewer for this valid point, and we have revised the figure to the best of our knowledge to improve the quality of the figure. We attempted to find out by extensive literature search about person correlation diagrams. Unfortunately, we did not find any.